# ARID1A Deficiency Regulates Anti-Tumor Immune Response in Esophageal Adenocarcinoma

**DOI:** 10.3390/cancers15225377

**Published:** 2023-11-12

**Authors:** Le Zhang, Yueyuan Zheng, Wenwen Chien, Benjamin Ziman, Sandrine Billet, H. Phillip Koeffler, De-Chen Lin, Neil A. Bhowmick

**Affiliations:** 1Department of Medicine, Cedars-Sinai Medical Center, Los Angeles, CA 90048, USA; le.zhang@cshs.org (L.Z.); zhengyy86@mail.sysu.edu.cn (Y.Z.); wen-wen.chien@cshs.org (W.C.); bziman@usc.edu (B.Z.); sandrine.billet@cshs.org (S.B.); h.koeffler@cshs.org (H.P.K.); 2Herman Ostrow School of Dentistry, University of Southern California, Los Angeles, CA 90033, USA

**Keywords:** ARID1A, esophageal adenocarcinoma, CD8^+^ T cells, IFN response, tumor immunity, lipid metabolism

## Abstract

**Simple Summary:**

It is known that 9–15% of esophageal adenocarcinoma (EAC) tumors have ARID1A mutations. Studies have shown that ARID1A mutations play a role in the tumor immune phenotype. We aimed to investigate the immunomodulating effects of ARID1A deficiency in EAC. Our study implicates that ARID1A plays a role in the regulation of both the immune response and lipid metabolism pathways. Our findings also provide valuable insights into the functional role of ARID1A in EAC and its potential implications for future clinical testing involving immune checkpoint blockade therapy. The identification of ARID1A as a potential biomarker for immune checkpoint blockade therapy may help in selecting patients who are more likely to benefit from this treatment approach.

**Abstract:**

ARID1A, a member of the chromatin remodeling SWI/SNF complex, is frequently lost in many cancer types, including esophageal adenocarcinoma (EAC). Here, we study the impact of ARID1A deficiency on the anti-tumor immune response in EAC. We find that EAC tumors with ARID1A mutations are associated with enhanced tumor-infiltrating CD8^+^ T cell levels. ARID1A-deficient EAC cells exhibit heightened IFN response signaling and promote CD8^+^ T cell recruitment and cytolytic activity. Moreover, we demonstrate that ARID1A regulates fatty acid metabolism genes in EAC, showing that fatty acid metabolism could also regulate CD8^+^ T cell recruitment and CD8^+^ T cell cytolytic activity in EAC cells. These results suggest that ARID1A deficiency shapes both tumor immunity and lipid metabolism in EAC, with significant implications for immune checkpoint blockade therapy in EAC.

## 1. Introduction

Recent trends in the incidence of esophageal cancer have highlighted the dramatic increase in esophageal adenocarcinoma (EAC) and a decrease in squamous cell carcinoma (ESCC) in Western countries [1]. The increase in EAC incidence is possibly due to both gastroesophageal reflux-associated Barrett’s esophagus and obesity-associated metabolic dysregulation [2].

The AT-rich interactive domain 1A (ARID1A) is a key subunit of the highly conserved chromatin remodeling complex SWI/SNF, which facilitates the accessibility of chromatin and promotes transcriptional activity [3]. ARID1A is an important tumor suppressor gene, one of the most frequently mutated in human cancers, including those of the ovary, endometrium, colon, bladder, pancreas and esophagus. Overall, around 6% of human cancers, including 9–15% of EAC tumors, have inactivation mutations of ARID1A [4,5,6,7,8]. The complete loss of protein expression of ARID1A occurs even when only one allele of ARID1A is mutated [9]. Interestingly, ARID1A mutations occur early in EAC carcinogenesis, and the frequency of ARID1A deficiency is elevated during the multistep progression of esophageal metaplasia–carcinoma [10]. 

Immune checkpoint blockade (ICB) therapy reinvigorates the anti-tumor cytotoxic activity of T cells, representing a significant advancement in cancer therapy [11]. However, only a limited fraction of patients demonstrate a clinical benefit. Thus, it is critical to identify potential patient candidates who are likely to benefit from ICB therapy. The level of tumor-infiltrating T cells, the tumor mutational burden, the expression of checkpoint molecules such as PD-L1 and LAG-3 and microsatellite instability are known factors impacting the clinical response to ICB therapy [12]. Tumor-infiltrating T cells are the determinants of immune tolerance and surveillance in tumors [13]. Studies have shown that ARID1A mutations play a role in the tumor immune phenotype, T cell immunity and the tumor immune microenvironment [14,15,16]. However, the functional implications of ARID1A mutations in the anti-tumor immune response in EAC remain incompletely understood. In this work, we focus on the immunomodulating effects of ARID1A deficiency in EAC. 

Obesity is considered one of the most important risk factors for EAC [17]. We have shown recently the association between a high-fat diet (HFD), obesity and EAC risk [18]. Intriguingly, accumulating lines of evidence suggest that obesity is associated with an improved response to ICB in diverse cancer types, including non-small cell lung cancer, melanoma, and renal cell carcinoma [19]. It was reported that ARID1A deficiency in hepatocytes promotes HFD-induced steatosis by impairing fatty acid oxidation, while wild-type ARID1A has a protective role against hepatic steatosis [20]. These results suggest that ARID1A plays a role in lipid metabolism. Palmitate is a common saturated long-chain fatty acid and a key component of HFD [21]. Based on these prior observations, we also explore the role of palmitate and ARID1A deficiency in the immune response against EAC tumor cells. 

## 2. Materials and Methods

### 2.1. Cell Lines 

Human EAC cells (FLO-1 and SKGT4) were obtained from ATCC and grown in DMEM medium supplemented with either 10% fetal bovine serum (FBS) or 3% charcoal-stripped FBS (for palmitic acid treatment where indicated) and 1% penicillin/streptomycin (all components from Thermo Fisher, Waltham, MA, USA). Murine gastric cancer cell lines YTN2, YTN5 and YTN16 were gifted by Sachiyo Nomura (University of Tokyo, Tokyo, Japan) and grown in DMEM medium, supplemented with 10% FBS, 1% L-glutamine, 0.5% penicillin/streptomycin and MITO (1:1000) on collagen I-coated plates. Moreover, 293T cells were obtained from ATCC. All cells were maintained in a 5% CO_2_ incubator at 37 °C with 5% relative humidity. Cells were tested monthly for mycoplasma (LT07118, Lonza, Rockland, ME, USA) and were clear of contamination.

### 2.2. Animal Studies

Ovalbumin (OVA)-specific T cell receptor transgenic OT-1 mice (6–8 week) were purchased from the Jackson Laboratory. All animal procedures were performed according to an approved protocol from the Institutional Animal Care and Use Committee at Cedars-Sinai Medical Center.

### 2.3. Lentivirus Infection

For shRNA-mediated gene silencing, pLKO.1-shRNA vectors (Sigma) were used for the knockdown of ARID1A: sh1 (GAAAGCGAGGGCCCCGCCGT), sh5 (GCTTCGGGCAACCCTACGGC). The target sequence for control scrambled shRNA was GAACCTATTCCCGCAATCTAA. For virus production, 293T cells were co-transfected with lentiviral packaging plasmids pSPAX2 (Addgene, 12260) and pMD2.G (Addgene, 12259) using Lipofectamine 3000. After 48 h, lentiviral particles were harvested and filtered through a 0.45 µM filter. EAC cells infected with lentiviruses were supplemented with 8 µg/mL polybrene and then selected by 2 µg/mL puromycin for 3–5 days.

### 2.4. RNA Preparation, cDNA Synthesis and qPCR

Total RNA was extracted using the RNeasy Plus Mini kit (Qiagen, Hilden, Germany) according to the manufacturer’s recommended protocol. A Nanodrop spectrophotometer (Thermo Fisher, Waltham, MA, USA) was used to measure RNA concentrations at 260/280 nm absorbance. Then, 500 ng of total RNA was used for cDNA synthesis with the iScript cDNA Synthesis Kit (Bio-Rad, Hercules, CA, USA). Quantitative real-time PCR reactions were performed using SYBR Green Mix (Azura Genomics, Raynham, MA, USA) in 96-well qPCR plates using the Thermos qPCR system (Thermo Fisher Quant Studio3), according to the manufacturer’s instructions. Data were calculated as mRNA expression relative to housekeeping genes (2^−ΔΔCt^). Results were obtained from at least three independent experiments and are shown as the mean ± SD. Primers were purchased from IDT (Coralville, IA, USA). Specific primer sequences can be found in Appendix A.

### 2.5. RNA Sequencing and Gene SET Enrichment Analysis

Total RNA was extracted with the RNeasy Plus Mini kit (Qiagen, Hilden, Germany) according to the manufacturer’s instructions. RNA library and transcriptome sequencing were performed by Quick Biology Inc. (Los Angeles, CA, USA). Gene set enrichment analysis (GSEA) of differentially expressed genes was performed by GSEA 4.0.3. 

### 2.6. Western Blot Assay

Extracted proteins were separated by 4–10% SDS-PAGE gels. The blots were probed with antibodies against ARID1A (12354S, Cell Signaling, Danvers, MA, USA) and β-actin (Cell Signaling, 4970), followed by incubation with alkaline phosphatase-conjugated secondary antibodies (A3687, Sigma-Aldrich, St Louis, MO, USA) for 1 h at room temperature. Blots were developed by 1-step NBT/BCIP (34042, Thermo Scientific, Waltham, MA, USA).

### 2.7. Murine CD8^+^ T Cell Isolation

Spleens from OT-1 mice were dissociated by nylon mesh, and red blood cells were lysed with RBC lysis buffer solution (4308151, eBioscience, San Diego, CA, USA). The CD8^+^ MojoSort Mouse CD8^+^ T cell isolation kit (480035, BioLegend, San Diego, CA, USA) was used to purify naïve CD8^+^ T cells from freshly harvested spleens. CD8^+^ T cells were then stimulated with 300ng/mL OVA peptide (Sigma-Aldrich, S7951) for 24 h, followed by removing OVA and changing to a medium containing IL-2 100 U/mL for 48 h. Cell purity was determined by flow cytometry (FACS) staining for CD3 and CD8. CD8^+^ T cells were further used for a cytotoxicity assay and transwell migration assay.

### 2.8. CD8^+^ T Cell Co-Culture Assay

Murine gastric cancer cells YTN5 and YTN16 were transfected with either siRNA against ARID1A or scramble siRNA and then incubated with 10 ng/mL IFN-γ for 24 h. Subsequently, they were co-cultured with CD8^+^ T cells at the ratio of 1:3 at 37 °C. After 24 h, the cells were fixed by methanol and stained with 0.5% crystal violet. The bound dye was resolved by 10% acetic acid (Sigma-Aldrich). The optical density of the solution at 595 nm was measured by the iMARK^TM^ Microplate Absorbance Reader (Bio-Rad, Hercules, CA, USA).

### 2.9. Transwell Migration Assay

CD8^+^ T cells were seeded into transwell inserts with 5 µm transwell filters. Culture media from YTN5 and YTN16 cells transfected with ARID1A siRNA or scramble siRNA, with 24 h 10 ng/mL IFN-γ incubation, were collected and added to the bottom chamber for 12 h. Migrated CD8^+^ T cells were counted by trypan blue. Transwell migration assays were performed independently at least three times.

### 2.10. Small Interfering RNA (siRNA) Transfection

YTN cells at 70% confluence were transfected with 25nM mouse ARID1A siRNA (L-040694-01, Dharmacon, TX, USA) and control scramble siRNA (sc-37007, Santa Cruz, TX, USA) using Lipofectamine 3000 (Fisher Scientific, Waltham, MA, USA), as described by the manufacturer. Then, 24 h after the addition of the transfection mix, liposomes were removed and fresh medium was added. The mouse ARID1A siRNA was a pool of 4 different siRNA target sequences: #1: AGAUGUGGGUGGACCGGUA;#2: AAGCAUUGCCCAAGAUCGA;#3: GGACAGGGGAUCAAUAGUA;#4: CCUUGGGGAUGUUAAGUUA.

### 2.11. Statistical Analysis

The comparison between different groups was performed by the paired Student’s *t* test. Two-way ANOVA was used to compare quantitative variable changes to the levels of two categorical variables. Results were expressed as the mean ± SD, and *p* < 0.05 was considered statistically significant. All the data were analyzed from at least 3 independent experiments. Graphs were prepared using the GraphPad Prism 6 software (GraphPad Software, San Diego, CA, USA).

## 3. Results

### 3.1. ARID1A Mutation Is Associated with High Tumor-Infiltrating CD8^+^ T Cell Levels in EAC Tumors

An initial correlative analysis of the inferred CD8^+^ T cell levels within tumor samples in the context of ARID1A mutations was performed using The Cancer Genome Atlas (TCGA) cohorts with the TIMER2.0 method [22,23,24,25]. Interestingly, EAC showed one of the most significant associations across all cancer types, wherein ARID1A mutations were correlated with greater CD8^+^ T cell infiltration as compared to ARID1A wild-type tumors (Figure 1A,B). We also found that ARID1A mutations were associated with the greater enrichment of the IFN-α and IFN-γ response pathways in EAC patients from both the Oesophageal Cancer Clinical and Molecular Stratification (OCCAMS) consortium (EGAD00010001822) and the TCGA database (Figure 1C).

We next explored these associations by generating ARID1A knockdown EAC cells (FLO1, SKGT4) in vitro. The isogenic silencing of ARID1A in FLO1 and SKGT4 cells was verified by Western blotting and quantitative PCR (Figure 2A,B, Appendix A). GSEA analysis of differentially expressed genes identified through RNA sequencing from ARID1A knockdown cells demonstrated the significant upregulation of the IFN-α and IFN-γ signaling pathways compared to the parental cells (Figure 1C), in agreement with EAC patient-derived data (Figure 1D). These data suggest that ARID1A mutation is associated with the IFN response pathway and may regulate the immune response in EAC cells.

### 3.2. Knockdown of ARID1A Promotes CD8^+^ T Cell Recruitment by IFN Signaling in EAC Cells

To test the role of ARID1A deficiency in the IFN response in EAC, we measured the expression of IFN-α and IFN-γ pathway genes in both shControl and shARID1A EAC cells. We observed a significant elevation in the expression of IFN pathway genes following ARID1A knockdown compared to control isogenic lines in both FLO1 and SKGT4 cells (Figure 2C,D). Among these genes, CXCL9, CXCL10 and BST2 were induced in both cell lines. Previous studies have shown that CXCL9 and CXCL10 are two critical cytokines that play a role in the recruitment of CD8^+^ T cells to the tumor microenvironment [26]. Since there are no murine EAC cell lines, we next utilized YTN cell lines, which were derived from gastric cancer models in C57BL/6 mice [27], for validation, considering that gastric cancer closely resembles EAC in both tumor biology and genomic landscape [28]. Accordingly, similar results were obtained in murine gastric cancer cell lines: ARID1A knockdown promoted the expression of IFN response genes (Figure 3). The expression of CXCL9 and CXCL10 was consistently induced by siARID1A in mouse cells.

To further investigate the relevance of ARID1A deficiency in the regulation of tumor-infiltrating CD8^+^ T cells in EAC, we performed a murine CD8^+^ T cell migration assay. The cell culture medium of murine cells was placed at the bottom chamber of the transwell plate. Freshly isolated murine spleen CD8^+^ T cells were seeded into the inserts. After 12 h, cells migrated into the bottom, and the medium was collected for cell counting (Figure 4A). We noted that the cell medium from the YTN5 and YTN16 lines with ARID1A silencing enhanced the migration of CD8^+^ T cells (Figure 4B,C). We next established an ex vivo co-culture platform to evaluate the antigen-dependent killing of tumor cells by CD8^+^ T cells (Figure 4D). Specifically, we pulsed YTN cells with ovalbumin peptide (SIINFEKL), an antigen that can be specifically recognized by OT-I CD8^+^ T cells via MHC-I [29]. Notably, we found that ARID1A knockdown cells were more sensitive to CD8^+^ T cell-mediated cytolytic activity in co-culture, indicating that ARID1A deficiency enhanced CD8^+^ T cell recruitment and the cell killing capacity (Figure 4E–H).

### 3.3. ARID1A Regulates Lipid Metabolism Genes in EAC

Our previous study elucidated a transcriptional regulatory loop of master regulator transcription factors, fatty acid synthesis and EAC risk [19]. Interestingly, ARID1A has also recently been implicated in the regulation of hepatic lipid metabolism [20]. This raises the question of whether ARID1A plays a role in regulating lipid metabolism in EAC cells. We observed that the expression of de novo lipogenesis genes, including FASN, SREBF1, MVD, SCD1 and SREBF2, was downregulated in ARID1A knockdown cells compared to the isogenic controls (Figure 5A,B; Appendix A). We also noted that cell proliferation was impaired by ARID1A knockdown in SKGT4 and FLO1 cells (Appendix A), consistent with reports in other cancers, wherein the loss of ARID1A led to reduced cell viability [30]. Furthermore, we performed gene expression correlation analysis between FASN and ARID1A using TCGA data, revealing that the expression levels of FASN and ARID1A were positively correlated in EAC (Appendix A). Both fatty acid synthesis genes, FASN and SCD1, were also significantly downregulated in SKGT4 cells upon ARID1A knockdown (Figure 5C).

It was reported that HFD could inhibit FASN expression in white adipose tissue, while a high-carbohydrate/low-fat diet induces FASN [31]. To model HFD conditions, we treated EAC cells with palmitate. Indeed, palmitate treatment inhibited FASN expression in FLO1 and SKGT4 cells, as well as SCD1 expression in FLO1 cells (Figure 5D,E). Consistently, there was the significant downregulation of FASN and SCD1 expression upon palmitate treatment in mouse cancer cells (Figure 5F). These data showed that FASN expression was downregulated by either ARID1A deficiency or palmitate treatment in EAC.

Given our present results implicating ARID1A in the regulation of both the immune response and lipid metabolism pathways, we explored whether palmitate could play a role in CD8^+^ T cell recruitment in EAC cells. We performed murine CD8^+^ T cell migration and co-culture killing assays in the presence or absence of palmitate. We found that palmitate treatment promoted CD8^+^ T cell recruitment (Figure 6A,B) and induced CD8^+^ T cell cytolytic activity compared to controls (Figure 6C,D). We also noted that CXCL9 and CXCL10 were significantly induced by palmitate treatment (Figure 4E,F). These results phenocopied the changes caused by ARID1A deficiency, suggesting that ARID1A may regulate the immune response pathways through the modulation of lipid metabolism.

## 4. Discussion

ARID1A is the most frequently mutated subunit of the SWI/SNF chromatin remodeling complex in cancer [32]. However, the functional relevance of ARID1A mutation is still incompletely understood in the context of EAC. In this work, we leveraged pan-cancer datasets from TCGA to discover the association between ARID1A genomic changes and CD8^+^ T cell infiltrates. Tumors with mutated ARID1A exhibited higher CD8^+^ T cell infiltration levels compared to those with wild-type ARID1A in EAC. In support of this, our RNA-seq data analysis showed that ARID1A knockdown correlated with an increased IFN signaling gene signature. We explored the potential impact of the ARID1A genetic status on the IFN signaling pathway and CD8^+^ T cell immunity in EAC in vitro. In line with our data, ARID1A mutation was found to be associated with increased immune activity in other gastrointestinal (GI) cancers, including the enrichment of CD8^+^ T cells, NK cells and immune-promoting M1 macrophages [33]. In addition, many studies have reported that ARID1A deficiency correlates with microsatellite instability (MSI), genomic features of GI cancers [15]. A high MSI status is known to confer high immunogenic activity in cancer and improve responsiveness to immunotherapy [16]. Indeed, a recent study showed that ARID1A-mutant cancer patients responded better to ICB treatment and had longer progression-free survival compared to ARID1A-wild-type patients [34]. These data suggest that mutations of ARID1A may regulate tumor immunity, and the ARID1A status may help to identify patients who are most likely to benefit from immunotherapies. Conversely, a recent work demonstrated that ARID1A mutations impair IFN signaling pathways, negatively shaping the tumor immune phenotype in ovarian cancer and colon cancer [14], indicating that the impact of ARID1A loss may depend on the tumor type. Further studies are warranted to elucidate the immunological role of ARID1A in the tumor microenvironment.

Research has shown that ARID1A regulates lipid metabolism [20,35]. The loss of ARID1A induces lipogenesis and inhibits fatty acid oxidation in hepatocellular carcinoma [36]. However, the role of ARID1A in lipid metabolism in EAC has not been established. Our findings revealed that ARID1A knockdown cells exhibited the downregulation of key genes involved in de novo lipogenesis, such as FASN, SREBF1, MVD, SCD1 and SREBF2. Indeed, the expression of FASN, a critical enzyme in fatty acid synthesis [37], was found to be positively correlated with ARID1A expression in EAC patients. FASN overexpression has been observed in many cancers and precancerous lesions [38]. It has been reported that the inhibition of FASN suppresses the proliferation, migration and invasion of various types of cancers [39,40]. The overexpression of FASN was also found in EAC and Barrett’s esophagus [41]. Consistently, the inhibition of FASN has been reported to reduce cell proliferation in EAC [42]. These data suggest that the lower proliferation rate of ARID1A-deficient cells might be partially due to FASN repression, which leads to decreased de novo lipogenesis [30].

Increased fat intake is associated with a higher body weight and BMI and the development of obesity [43]. Recent epidemiological evidence demonstrates that obesity is a prognostic factor for ICB therapy [42]. A high body mass index (BMI) is positively associated with responsiveness to anti PD-1/PD-L1 treatment in patients with various cancers, such as melanoma, renal cell carcinoma and non-small cell lung cancer [44]. Moreover, studies have shown that enhanced fatty acid catabolism is important in maintaining CD8^+^ T cell function and increasing the efficacy of ICB therapy in melanoma [45]. Congruently, we observed that palmitate enhanced CD8^+^ T cell recruitment and cytolytic activity. In addition, CXCL9 and CXCL10 expression was significantly induced by palmitate treatment. These findings suggest a potential link between ARID1A, lipid metabolism and immune response pathways in EAC.

Our study has important implications for the understanding of ARID1A deficiency in the context of tumor immunity and lipid metabolism. Our findings provide valuable insights into the functional role of ARID1A in EAC and its potential implications for future clinical testing involving ICB therapy. The identification of ARID1A as a potential biomarker for ICB therapy may help to select patients who are more likely to benefit from this treatment approach. Furthermore, the link between ARID1A, lipid metabolism and immune response pathways supports the targeting of lipid metabolism as a potential therapeutic strategy for this deadly cancer.

## 5. Conclusions

The present study indicates that ARID1A could regulate tumor immunity in esophageal adenocarcinoma, and the ARID1A status may help to identify patients who are most likely to benefit from immunotherapies. Our findings also suggest a potential link between ARID1A, lipid metabolism and immune response pathways in EAC, indicating that lipid metabolism could be a potential therapeutic target.

## Figures and Tables

**Figure 1 cancers-15-05377-f001:**
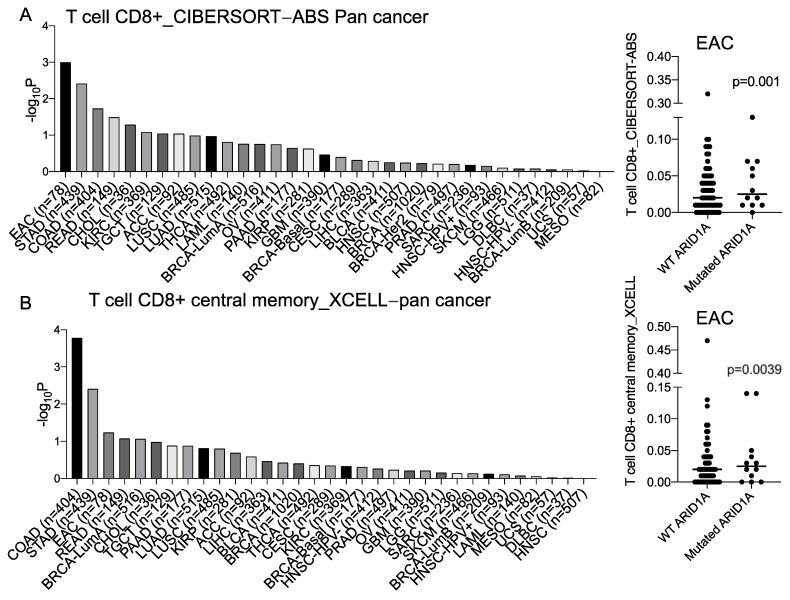
ARID1A mutations display increased tumor-infiltrating CD8^+^ T cells and IFN signaling pathways in EAC. (**A**,**B**) The statistical difference in tumor-infiltrating CD8^+^ T cells between mutated ARID1A and unmutated ARID1A in pan− cancer by (**A**) CIBERSORT− ABS and (**B**) XCELL. (**C**) The GSEA plot shows the positive enrichment of IFN-α and IFN-γ responses in the ARID1A knockdown cells compared to controls by shRNA in FLO1. (**D**) Volcano plots show GSEA results of the enriched IFN pathways in ARID1A-mutated compared to unmutated samples from OCCAMS and TCGA. Abbreviations: STAD: stomach adenocarcinoma, COAD: colon adenocarcinoma, READ: rectum adenocarcinoma, CHOL: cholangiocarcinoma, KIRC: kidney renal clear cell carcinoma, TGCT: testicular germ cell tumors, ACC: adrenocortical carcinoma, LUSC: lung squamous cell carcinoma, LUAD: lung adenocarcinoma, THCA: thyroid carcinoma, LAML: acute myeloid leukemia, BRCA: breast invasive carcinoma, OV: ovarian serous cystadenocarcinoma, PAAD: pancreatic adenocarcinoma, KIRP: kidney renal papillary cell carcinoma, GBM: glioblastoma multiforme, CESC: cervical squamous cell carcinoma and endocervical adenocarcinoma, LIHC: liver hepatocellular carcinoma, BLCA: bladder urothelial carcinoma, HNSC: head and neck squamous cell carcinoma, PRAD: prostate adenocarcinoma, SARC: sarcoma, SKCM: skin cutaneous melanoma, LGG: brain lower-grade glioma, DLBC: lymphoid neoplasm diffuse large B cell lymphoma, UCS: uterine carcinosarcoma, MESO: mesothelioma.

**Figure 2 cancers-15-05377-f002:**
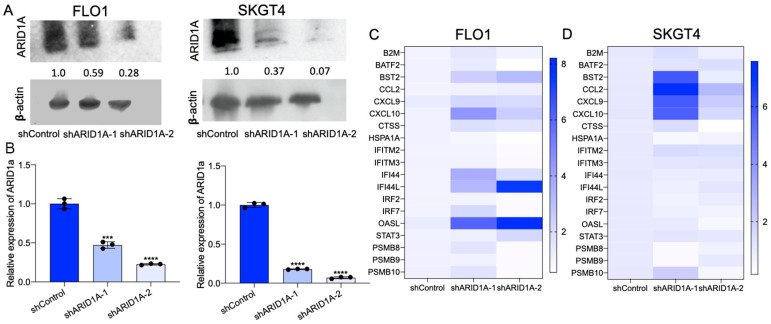
ARID1A knockdown induces IFN signaling response in EAC. (**A**) Western blot and (**B**) real-time PCR of ARID1A in FLO1 and SKGT4 ARID1A knockdown cells by shRNA. Representative blots are shown with the mean relative quantitation indicated normalized to ß-actin expression. (**C**) The expression of IFN signaling response-related genes in FLO1 and SKGT4 ARID1A knockdown cells by shRNA. *n* = 3. Paired, 2-tailed *t* test: *** *p* < 0.001, **** *p* < 0.0001. *p* values for (**C**,**D**) were determined by one-way ANOVA.

**Figure 3 cancers-15-05377-f003:**
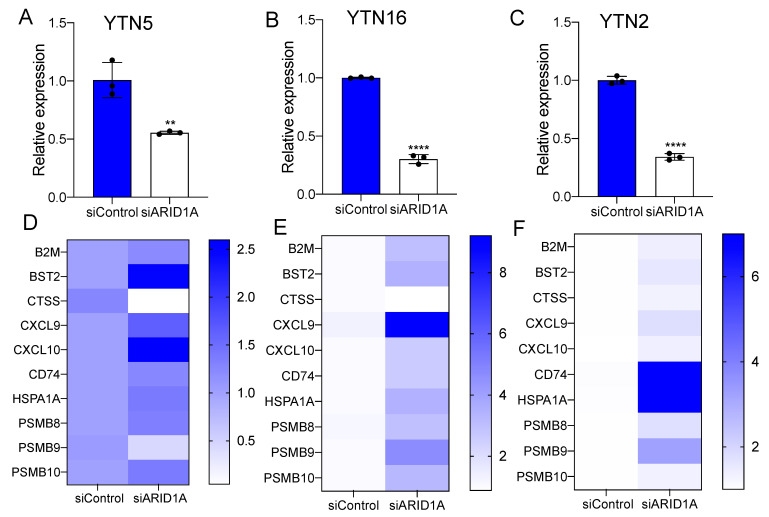
ARID1A knockdown induces IFN signaling response in murine EAC. (**A**–**C**) The mRNA expression level of ARID1A in ARID1A knockdown by siRNA in (**A**) YTN5, (**B**) YTN16 and (**C**) YTN2. (**D**,**F**) The expression of IFN signaling response-related genes in (**D**) YTN5, (**E**) YTN16 and (**F**) YTN2 transfected with ARID1A siRNA. *n* = 3. Paired, 2-tailed *t* test: ** *p* < 0.01, **** *p* < 0.0001. *p* values for (**D**,**F**) were determined by one-way ANOVA.

**Figure 4 cancers-15-05377-f004:**
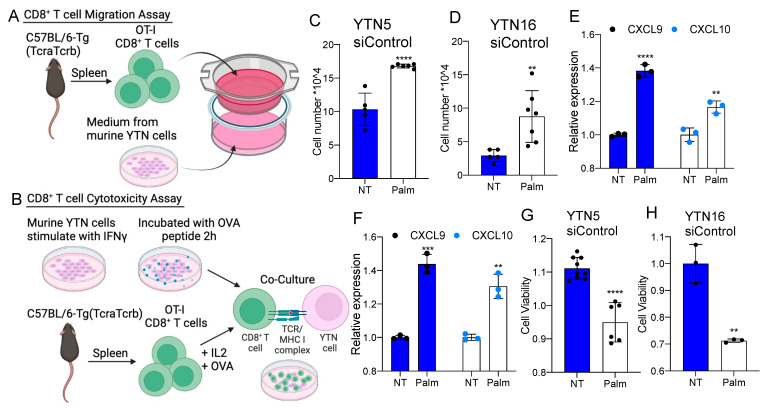
Palmitate potentiates CD8^+^ T cell recruitment and cytotoxicity in mouse EAC. (**A**) A diagram of CD8^+^ T cell migration assay. (**B**) A diagram of CD8^+^ T cell cytotoxicity co-culture assay. (**C**,**D**) The number of CD8^+^ T cells migrated into the culture medium from siControl of (**C**) YTN5 and (**D**) YTN16 with and without 48 h palmitate treatment. (**E**,**F**) The mRNA expression of CXCL9 and CXCL10 in siControl of (**E**) YTN5 and (**F**) YTN16 with and without 48 h palmitate treatment. (**G**,**H**) The cell viability of (**G**) YTN5 and (**H**) YTN16 siControl with and without palmitate treatment after CD8^+^ T cell cytotoxic killing for 24 h. *n* ≥ 3. Paired, 2-tailed *t* test: ** *p* < 0.01, *** *p* < 0.001, **** *p* < 0.0001.

**Figure 5 cancers-15-05377-f005:**
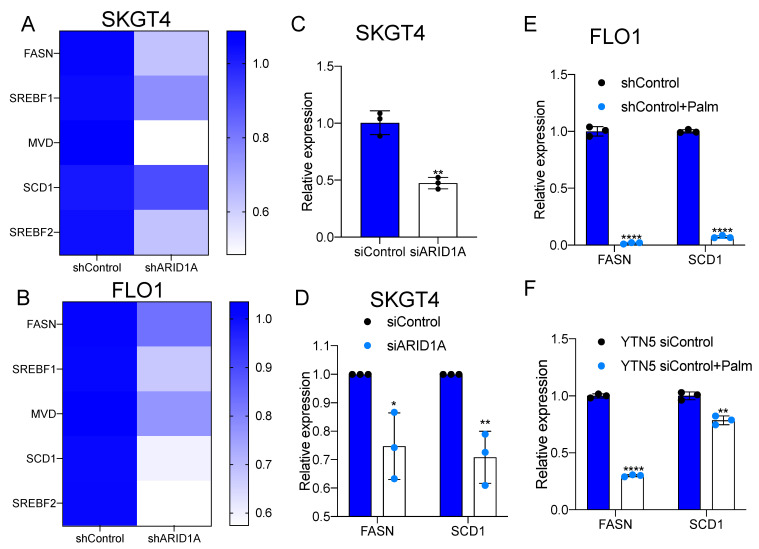
ARID1A knockdown inhibits de novo lipogenesis in EAC. (**A**,**B**) The expression of de novo lipogenesis-related genes in (**A**) SKGT4 and (**B**) FLO1 under ARID1A knockdown by shRNA. (**C**) The mRNA expression under ARID1A knockdown by siRNA in SKGT4. (**D**) The expression levels of FASN and SCD1 in SKGT4 transfected with ARID1A siRNA. (**E**,**F**) The mRNA expression of FASN and SCD1 in (**E**) FLO1 and (**F**) YTN5 control isogenic cells generated for ARID1A knockdown EAC treated with palmitate for 48 h. *n* = 3. Paired, 2-tailed *t* test: * *p* < 0.05, ** *p* < 0.01, **** *p* < 0.0001. *p* values for (**A**,**B**) were determined by one-way ANOVA.

**Figure 6 cancers-15-05377-f006:**
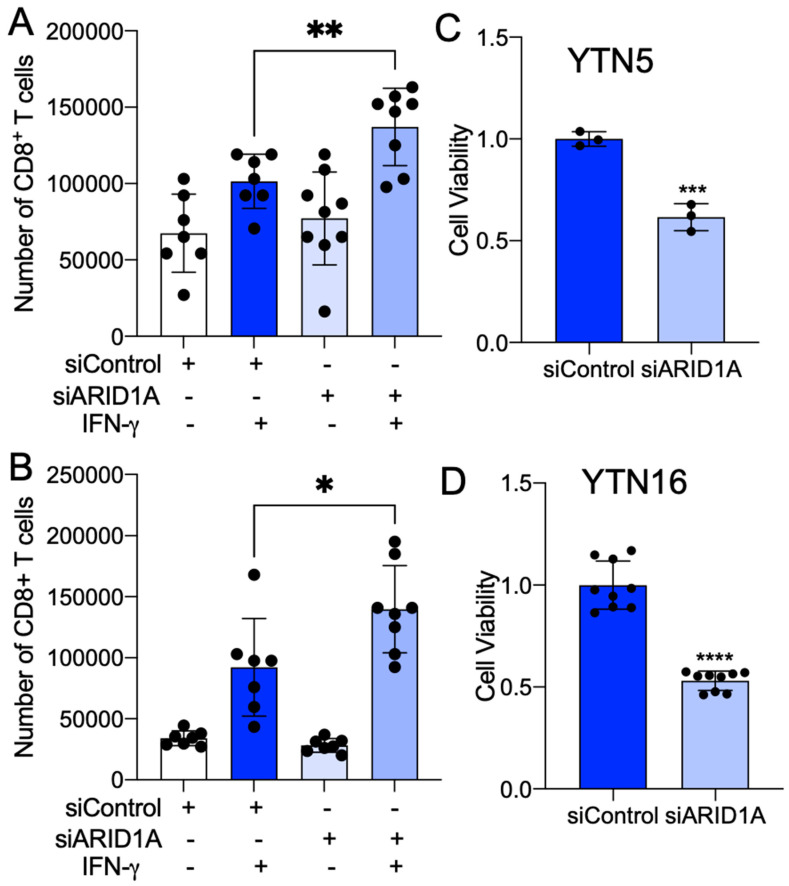
ARID1A knockdown promotes CD8^+^ T cell recruitment and cytotoxicity in mouse EAC. (**A**,**B**) The absorbance at 595 nm for the viable cells of (**A**) YTN5 and (**B**) YTN16 with and without ARID1A knockdown by siRNA after CD8^+^ T cell cytotoxic killing for 24 h. (**C**,**D**) The number of CD8^+^ T cells migrated into the culture medium from (**C**) YTN5 and (**D**) YTN16 with and without ARID1A knockdown by siRNA. *n* ≥ 3. Paired, 2-tailed *t* test: * *p* < 0.05, ** *p* < 0.01, *** *p* < 0.001, **** *p* < 0.0001.

## Data Availability

The data presented in this study are available in this article.

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
