# Peer review of "ARID1A Deficiency Regulates Anti-Tumor Immune Response in Esophageal Adenocarcinoma"

_cancers, 2023, doi:10.3390/cancers15225377_

Round 1

Reviewer 1 Report

Comments and Suggestions for Authors

The manuscript entitled ‘ARID1A deficiency regulates anti-tumor immune response in esophageal adenocarcinomawas well received. In current study the authors have tried to demonstrate that ARD1A increased IFN-γ responsiveness in esophageal adenocarcinoma. It was revealed that ARD1A may regulate lipid metabolism and ultimately affect T cells recruitment to the tumor site. Here are some concerns.

State clearly the novelty of this study?

The fact that ARD1A can promote T cell infiltration and sensitize IFN-γ pathway has already been published. What the current results add to the already known data with reference to these publications specifically. https://doi.org/10.3390/cells8070678 and https://doi.org/10.1172/JCI134402 .

ARD1A deficiency may regulate MHC-1 expression. It can also upregulate PDL-1. As both these are pivotal for CD8 T cell mediated immune responses, I strongly recommend determining these in respective cell lines.

The authors stated that the addition of culture medium from cell lines promoted T cell migration? By what mechanism this happened.? Later on it was revealed that palmitate treatment promoted CD8 T cell recruitment. Why not palmitate not quantified in the culture medium? The authors must quantify palmitic acid in the culture medium to validate the conclusion withdrawn.

Provide original blot uncropped gel images for WB. 

Author Response

Thank you for your review. Please see the attached file. Thank you again!

Reviewer 2 Report

Comments and Suggestions for Authors

Zhang et al. have conducted a study finding that ADID1A mutations are associated with a higher level of tumor-infiltration CD8+ T cells in EAC and ARID1A deficiency in lipid metabolism in EAC. This work could illustrate the role of ARID1A as a biomarker for immune checkpoint inhibitors. There are a few questions and concerns that the authors could address before the acceptance of this work.

Major

1)    What is the association between ARID1A mutations and ARID1A knockdown or deficiency? The cancer genomic database indicated a correlation between ARID1A mutation and CD8+ T cell infiltration, but the experiment focused on the knockdown of ARID1A and the downstream effect. Is there evidence that the mutations in ARID1A lead to reduced transcription or translation? Why did the authors knockdown ARID1A in the in-vitro models instead of introducing mutant ARID1A expression in the in-vitro models?

2)    The western blot is a concern. First, please include protein ladders on the membrane; second, there seems to be a smear of proteins or significant unspecific binding of the antibody of ARID1A. Please refine the blocking process or antibody specificity. Third, the housekeeping gene is missing in the SKGT4 cell line. What is the reason behind that?

3)    The cell culture medium contains no fatty acid supplement, which means the cells could not access excessive lipids to promote proliferation or migration, and the transcriptional profile may not reflect actual tumor activities in a high lipid environment. The authors should consider supplementing fatty acids in the culture medium and study whether the knockdown of ARID1A affects the utilization of lipids in tumor cells and their proliferation or migration activity. Staining of intracellular lipid accumulation could be considered as well.

Minor

1)    Please include abbreviations for TCGA tumor types in Figure 1 and clarify that STAD, as stomach adenocarcinoma, has the most similar correlation between ARID1A mutation and T-cell infiltration, which is why the authors used gastric cell lines to substitute the lack of murine EAC cell lines.

2)    Please show data dots on all bar graphs and indicate the N number in figure legends and methods.

Author Response

Thank you for your review! Please see the attached file. Thank you again !

Round 2

Reviewer 1 Report

Comments and Suggestions for Authors

The response seems satisfactory and the paper can be accepted

Reviewer 2 Report

Comments and Suggestions for Authors

Concerns have been addressed, no further comments.